# A Framework for Continuous Authentication Based on Touch Dynamics Biometrics for Mobile Banking Applications

**DOI:** 10.3390/s21124212

**Published:** 2021-06-19

**Authors:** Priscila Morais Argôlo Bonfim Estrela, Robson de Oliveira Albuquerque, Dino Macedo Amaral, William Ferreira Giozza, Rafael Timóteo de Sousa Júnior

**Affiliations:** Cybersecurity INCT Unit 6, Decision Technologies Laboratory—LATITUDE, Electrical Engineering Department (ENE), Technology College, University of Brasília (UnB), Brasília 70910-900, Brazil; robson@redes.unb.br (R.d.O.A.); dinoamaral@gmail.com (D.M.A.); giozza@unb.br (W.F.G.); desousa@unb.br (R.T.d.S.J.)

**Keywords:** continuous authentication, touch dynamics biometrics, mobile security, mobile authentication

## Abstract

As smart devices have become commonly used to access internet banking applications, these devices constitute appealing targets for fraudsters. Impersonation attacks are an essential concern for internet banking providers. Therefore, user authentication countermeasures based on biometrics, whether physiological or behavioral, have been developed, including those based on touch dynamics biometrics. These measures take into account the unique behavior of a person when interacting with touchscreen devices, thus hindering identitification fraud because it is hard to impersonate natural user behaviors. Behavioral biometric measures also balance security and usability because they are important for human interfaces, thus requiring a measurement process that may be transparent to the user. This paper proposes an improvement to Biotouch, a supervised Machine Learning-based framework for continuous user authentication. The contributions of the proposal comprise the utilization of multiple scopes to create more resilient reasoning models and their respective datasets for the improved Biotouch framework. Another contribution highlighted is the testing of these models to evaluate the imposter False Acceptance Error (FAR). This proposal also improves the flow of data and computation within the improved framework. An evaluation of the multiple scope model proposed provides results between 90.68% and 97.05% for the harmonic mean between recall and precision (F1 Score). The percentages of unduly authenticated imposters and errors of legitimate user rejection (Equal Error Rate (EER)) are between 9.85% and 1.88% for static verification, login, user dynamics, and post-login. These results indicate the feasibility of the continuous multiple-scope authentication framework proposed as an effective layer of security for banking applications, eventually operating jointly with conventional measures such as password-based authentication.

## 1. Introduction

Currently, at least 5 billion people use mobile telephones [1], including 3.2 billion people who use smartphones [2], and among them, 2 billion use their smartphones to access banking applications [3]. With the widespread adoption of these devices, there is a related growing number of malware specifically targeting mobile devices. As reported in a Kaspersky lab report released in 2019, the number of attacks on mobile devices doubled in 2018, with more than 1165 million [4]. This migration to mobile applications motivated the evolution of authentication methods over time, aiming to ensure fraud prevention, especially in the case of critical applications such as financial ones.

Commonly, the first interaction of a user with a mobile device and applications is the authentication process. There are three traditional methods to authenticate a user: possession of something, knowledge of information, and biometrics, i.e., something that is part of the person’s body or behavior [5]. Biometric authentication is an interesting means of hindering fraud because of the effort required to forge something that is part of a person compared to the effort related to producing something that the person knows or owns.

Biometric authentication can be classified according to two operational forms [6]: physiological authentication, which is linked to the person’s physical characteristics, such as irises or fingerprints, and behavioral authentication, which is linked to attitudes that are inherent to the individuals, e.g., the way they walk or how they interact with a touchscreen device.

Password-based authentication is the most commonly used method to protect data from intruders [7], only requiring user authentication in ordinary situations through a password or another credential. Usually, this occurs for the first interaction with the application during login. One of the possibilities to increase the security of authentication processes is to authenticate the user during the entire interaction with an application and not only during login. Therefore, continuous authentication approaches can provide an additional line of defense, especially if designed as a non-intrusive and passive security countermeasure [8], as it can occur implicitly and transparently, providing an interesting balance between security and usability.

Thus, implicit authentication schemes use behavioral biometrics to authenticate the identity of the user continuously and transparently [9]. For example, behavioral biometrics can be captured through user interaction with a smartphone’s touchscreen, generating features that uniquely identify the user. Recent studies revealed that smartphone sensors have a rich potential to be used in active/continuous authentication [10].

Considering this scenario, this paper proposes an improvement to the performance of the *Biotouch* framework, extending a previous related work [11]. This referenced work proposed a supervised machine-learning-based multimodal framework for continuous and implicit authentication. It was based on biometric behavior patterns obtained from mobile banking applications. These patterns are observed during continuous interaction of the user with the mobile application. The model uses data from the touchscreen interaction, the location captured via GPS, and information collected from various sensors. Such information allows us to detect possible fraudsters, thus generating alerts in the case of possible malicious behavior. The results obtained in this previous version of the *Biotouch* framework showed an accuracy between 78% to 91%, take into consideration only one scope and the rate of imposters was not considered.

The new, enhanced model proposed in this paper uses more than one scope and tests its resilience against imposters. Hence, using the new, enhanced model supported by multiple scopes for continuous authentication based on biometric authentication factors, the results show a new, effective, and high-performance model, with the accuracy varying between 82.53% and 96%. Regarding the percentage of errors, the possible number of impostors accepted, and the number of legitimate users rejected using Equal Error Rate (EER) metrics, the results show a variation between 0 and 11.5% when considering the best results obtained. Additionally, using multiple scopes, the performance of the *Biotouch* framework shows the harmonic mean between recall and precision (F1 Score (F1)), with results between 90.68% and 97.05%. The imposter-acceptance error rate and legitimate-user rejection rate, using the EER results, are between 9.85% and 1.88% for static verification considering login, post-login, and user dynamics. These results show the feasibility of using the proposed system as another layer of security for mobile banking applications, operating in conjunction with other traditional security methods, such as passwords, since the methods can coexist and enhance the security of mobile systems against fraud.

### 1.1. Main Contributions of This Work

The main contributions of this paper are as follows:Multiple-scope approach is uesd so that the authentication models are validated for different feature sets, with the best performing scopes added to the framework.Six different scopes were developed to improve the performance compared to the results previously presented in [11], which used only one scope.With the selected scopes, the efficiency metric of the models presents a minimum F1 score of 90%.Although the proposal validation uses templates of all users who participated in the data collection scenarios, only those models presenting a FAR below 10% were integrated into the new framework.

Considering this set of contributions, our results show that the new models created for the enhanced *Biotouch* framework provide higher resiliency when compared with the previous results achieved in [11].

### 1.2. Organization of This Work

This paper is organized as follows. Section 2 summarizes the related works and some considerations about their results. In Section 3, the improvements of the continuous authentication framework are discussed. Section 4 presents and discusses the validation results of the experiments with the new proposed framework. Lastly, Section 5 provides the conclusion and proposes future lines of work.

## 2. State-of-the-Art

In this section, a literature review describing the main concepts that support the development of the framework is presented. Additionally, the discussion of related work leads to a definition of the features used and a benchmark for the comparison of performance indicators for the proposed framework. It is also considered that a touch dynamics authentication framework must be composed of three phases: user enrollment, user authentication, and data retraining [12].

### 2.1. Security and Mobile Banking Applications

Online banking systems require efficient security models capable of identifying users and authorizing transactions, thus mitigating fraud [13]. As smart devices have become commonly used to access internet banking applications, these devices constitute appealing targets for fraudsters. Impersonation attacks are an essential concern for internet banking providers. In this context, the main challenge for electronic banking is ensuring the correct usage and verification of applications for banking security.

Usually the model for common attacks against online banking systems is to exploit vulnerabilities inherent in the people (engineering social and phishing) and then to gain control of the device (malware) and to steal the credentials of a legitimate user (fake Web pages and malware) [13]. Therefore, user authentication countermeasures based on biometrics have been developed, including those based on touch dynamics biometrics. Biometric characteristics are specific to the user and difficult to copy, share, and distribute [14].

### 2.2. Continuous Authentication for Mobile Banking Applications Based on Touch Dynamics Biometrics

Touch dynamics biometrics refers to the process of measuring and assessing human touch actions on the touchscreen of mobile devices [12]. To characterize the biometric behavior when using a touchscreen device, an individual’s biometric pattern is modeled based on information collected from the various sensors that make up modern smartphones, such as the accelerometer, ambient light sensor, typing compass, gyroscope, GPS, proximity sensor, touchscreen, and Wi-Fi [15].

In the context of mobile applications, behavioral biometrics emerged as a less intrusive biometrics model that can be captured implicitly. Additionally, it provides a greater balance between security and usability, especially because the touch behavior (information captured) is not part of the user’s private information.

Active and continuous authentication can be defined as the continuous verification of a person’s identity based on aspects of their behavior when interacting with a computing device [16]. The main characteristic of an authentication method is continuity, as authentication is constant during the entire time that the user interacts with the device by means of re-authentication tasks that occur periodically and transparently. The entire authentication process can be performed in the background without interrupting the user’s activities [17].

Behavioral biometrics, based on touch dynamics for authentication applications, have been investigated in recent years. An experiment regarding data collection for authentication is presented in [7], aimed at continuous static verification, with a sample of 42 users. The devices used in the cited experiment were a Nexus 7 tablet and a Mobil LG Optimus L7 II, and the users needed to enter the same password (considered strong) 30 times in each session. In this work, the supervised machine learning algorithm that presented the best performance was Random Forest (RF), which resulted in 82.53% and 93.04% accuracies, respectively, for the sets of 41 and 71 selected characteristics.

In [8], an experiment for capturing gestures was proposed for users, where each guest interacted with an application by reading three documents about different subjects and by interacting with two images to find the differences. The purpose was to collect data from the interaction of users with a touchscreen, including horizontal and vertical sliding, through continuous dynamic verifications. Forty-one volunteers participated in this research, using five different smartphones: Droid Incr., Nexus One, Nexus S, Galaxy S, and all on Android 2.3.x. Using the supervised machine learning algorithm Support Vector Machine (SVM), the best performance results for EER were 0% for intra-sessions, 2–3% for inter-sessions, and 4% when the authentication occurred a week after enrolment.

More recent work specifically focused on banking applications in order to increase the spectrum of fraud identification. For instance, the authors of [18] developed a banking analogous application with continuous static verification by considering password typing and by evaluating it on a set with 95 participating volunteers and data captured from the touch interaction and sensors available on smartphones, obtaining a 96% accuracy with the RF algorithm. In [19], using a fuzzy-based classifier, a static and dynamic continuous authentication model was proposed with the data captured from touchscreen and accelerometer interactions in an application developed using the characteristics of a real mobile banking application and from use by 22 volunteers, giving an EER of 11.5%.

### 2.3. Location and Continuous Authentication

The inclusion of information from a user in continuous authentication processes can contribute as an additional factor in verifying a user’s usage pattern for a mobile application. Location is one contextual information that is, for instance, used in works such as [16,20] as one of the factors that make up so-called multimodal authentication schemes for mobile applications.

In [16], the user’s location data were used in conjunction with the user’s movement information and device usage. Two profiles were established for each user: one for weekdays and the other for weekends. The pattern was set based on each user’s history, and the K-means algorithm was used to group the location data. In [20], the user’s location pattern data were used in a proposal for a multimodal biometrics system that combined GPS information, stylometry, the use of the application, and the web search pattern.

As location constitutes information that generally remains consistent in a user’s pattern of use of mobile applications, this contextual information contributes as additional data to validating a user’s pattern in conjunction with other factors.

### 2.4. Data Fusion

In this paper, fusion is an approach used to combine data and information from multiple sources to improve the accuracy or performance of a biometric authentication method. The information of various sources can be combined in four different ways: image-level fusion, scoring-level fusion [12], decision-level fusion, and feature-level fusion [21].

In multimodal authentication frameworks, one of the issues that need to be resolved is how to merge the classification results obtained for each of the used modalities. To solve this issue, the work in [16] used the Fusion Decision Center technique, which collects decisions from a local detector and uses them to define whether the result is −1 or 1. The scoring approach can also be used for merging results, as noted in [22], where a decision center combines all of the scores of the modalities, generating an overall decision score.

## 3. Proposed Model

As mentioned before, this work improves the *Biotouch* framework [11], seeking better performance and robustness by using a new union of scopes and by adding a new step in the authentication process, this latter being defined as the imposters’ FAR (I_FAR) threshold. The new, proposed framework model aims to capture the features of a user’s interaction with a mobile application via touchscreen. The model considers two main verifications: one static and the other dynamic. Static verification (SV) is achieved when a user types their password at login, while dynamic verification (DV) runs when the user interacts with the application after login.

The main objectives in the new version of the *Biotouch* authentication framework are to identify which of the proposed scopes, among six in the new version, perform better and to detect whether these scopes can be combined to generate better results than the previous ones found in [11]. The new scope results are checked based on the F1 score metric to validate which scopes are complementary to each other. Other goals include investigating if the inclusion of the new FAR test step for imposters makes the model more robust and verifying if the supervised machine learning algorithms that present the best results in [11] can be kept. These validations are important to understand the improvements brought to the quality of the authentication models in the new version of the *Biotouch* framework. Additionally, in order to investigate how the use of touch dynamics biometrics can reduce fraud in banking applications, we conducted an experiment with an application developed with characteristics similar to that of a real banking application. This application requires few touches with a short period of interaction with the user, but it serves the purposes of evaluation because, when such types of applications are attacked, it usually involves the loss of finances.

### 3.1. The New Framework Description

The proposed model involves checking the patterns of typing and sliding, and the location. This model is defined to cover both the login time, named Moment 1, and the interaction with the application, named Moment 2. The framework is illustrated in Figure 1. The two moments of user authentication are characterized differently:Moment 1: typing password, is classified as SV;Moment 2: interaction with the application to carry out a transaction is classified as DV.

The rule for merging the classifier results is based on score, using the accuracy (AC) of the location and the F1 score for the password-typing pattern. These characteristics are considered SV. For the application’s pattern interaction, it is considered DV.

Extending [11], one more step is included, i.e., the I_FAR check, and different from this previous work, the evaluation metric is replaced by the F1 score, which is applied for both SV and DV to cover all of the new framework scopes. Thus, referencing Figure 1, the detailed steps of the new authentication framework are the following:Step 1: Capture the pattern of location and password typing;Step 2: Calculate AC for the location pattern captured in Step 1 using the best model (which obtained AC ≥ 90% in the tests that are described hereafter), and calculate the F1 Score for the password-typing pattern captured in step 1 using the best model (which obtained F1 ≥ 90% and I_FAR ≤ 10% in the tests);Step 3: Fuse the AC for the location pattern with the F1 Score for the password-typing pattern using the simple arithmetic mean.Step 4: If the result of Step 3 is a score below 90%, an alert is generated, indicating a possible imposter;Step 5: Capture the location and interaction pattern with the application post-login activities;Step 6: Calculate the accuracy (AC) for the location pattern captured in Step 5 using the model that obtained AC ≥ 90%, and calculate F1 for the pattern of post-login interaction with the application in Step 5 using the best model that obtained F1 ≥ 90% and I_FAR ≤ 10%;Step 7: Fuse the AC, for the location pattern, with the F1 Score, for the pattern interaction with the application post-login activity, using the simple arithmetic mean;Step 8: If the result of Step 7 is a score below 90%, an alert is generated, indicating a possible imposter.

The following subsections describe the methods used to create the models mentioned in these steps.

### 3.2. Data Collection

For data collection, an Android application named *Biotouch* was developed and published on the Play Store. The application consists of a registration screen, a login screen (L), a menu service screen (MS), and two more screens for each of the three available services: (a) account screen (Cc), one menu account screen (Cc1) and one from the account transaction screen (Cc2); (b) transfer screen (T), one transfer menu screen (T1) and one transfer transaction screen (T2); and (c) payment screen (P), one for payment menu screen (P1) and one for the payment transaction screen (P2).

The number of people that participated as volunteers and attended the experiment were 51. The collection period was two weeks. The number of generated templates ranged from 9 to 630 between the users, depending on how many times the user interacted with the app screen. It represents a total of 3443 templates, used in the experiments, composed of various data vectors, corresponding to each touch of the user on the screen. The templates were saved on the Firebase platform. The registration flow is detailed in Figure 2, and the service flow is detailed in Figure 3.

To start using the *Biotouch* application, the user needs to register a password with 6 to 8 numeric digits. The user identifier is transparently defined by the installation task, and no action from the user is necessary to inform an identifier at the time of registration since the Firebase platform provides an Instance ID that is used as a unique identifier for each instance of the application [23]. Thus, this Instance ID is used as the user’s unique identifier.

During data collection, users were asked to interact at least five times a day with the application during the experiment period, no matter which of the flows were executed. The user should interact with five different screens to complete each selected flow.

Each usage template is then represented by the events generated by the users’ touch made during an interaction with each screen. Therefore, the number of vector data used for each authentication can vary.

The smartphone models used in the experiment were SM-G973F, SM-G9600, LG-M250, Moto G (4), SM- A305GT, SM-G9650, SM-G970F, SM-G975F, ASUS X00QD, F670S, GM1900, GT-I9500, LGM-M700, MI 8, Mi 9T, Mi A2, Mi A3, Moto E(4), Moto X4, Moto Z2 Play, MotoG3, One Vision, POCOPHONE F1, Redmi 7, Redmi Note 4, Redmi Pro, SM-A530F, SM-G530H, SM-G570M, SM-G930F, SM- G935F, SM-G955F, SM-G955U, SM-N950F, SM-N9600, SM-N970F, SM-N975F, X00HD, X00LD, XT1635-02, and XT1710-02.

The Android versions on the smartphones were 5.0.1, 5.0.2, 6.0, 6.0.1, 7.0, 7.1.1, 8.0.0, 8.1.0, 9, and 10.

About the weakness or attack surface of the captured templates of the users, the data were sent to the Firebase platform using HTTPS with the latest version of TLS. During the collection period, the data were available in memory and could only be captured if an attacker had access to the device’s memory.

### 3.3. Selection of Features

The extraction of touch dynamics biometrics features (catches) could be performed in different ways: spatial, movement [12], temporal, dynamic, and geometric [17]. In the proposed framework, feature generation was performed with the data collected by various sensors: accelerometer, gyroscope, magnetometer, orientation, linear acceleration, and gravity. Additionally, information regarding the pattern of interaction with the touchscreen comprised different ways of extracting touch biometrics, as detailed in Table 1.

Overall, a total of 29 features were obtained in Moment 1 and 31 features were obtained in Moment 2. Two features, the coordinates X and Y, and the latitude and longitude that are used to define the location pattern were added to the latter. These features extend the features used in [24], considering all features generated by sensors declared in related works, and adds features generated by more two sensors: rotation and acceleration.

The representation of the X, Y, and Z-axes on a smartphone is detailed in Figure 4 to facilitate the understanding of the collection of features about axes as detailed in Table 1.

For the creation of a feature ranking for Moments 1 and 2, the Random Forest (RF) algorithm was used with an impurity criterion based on entropy and a multiclass model. All user templates considered the hyperparameters defined in the Table 2 that were defined empirically based on the best results obtained during training and considering the test time.

### 3.4. Model Creation and Parameter Test

According to [28], it is not an easy task to find a machine learning classifier suitable for user authentication. This is the reason we evaluate several algorithms in this paper. The Random Forest (RF) algorithm [29,30,31] was chosen based on the good performance presented in [24]. Two more algorithms based on ensemble methods, Gradient Boost (GB) [32,33] and Extreme Gradient Boosting (XGB) [34], were also selected. These are considered more modern algorithms with high performance, despite requiring greater computational power for training. Oppositely, two algorithms based on probabilities, i.e., the Naive Bayes Bernoulli (NBB) and Naive Bayes Gaussian (NBG) [31,35,36], were also added because they are simpler, thus implying low processing costs while being rapid for prediction and training. One more algorithm, Support Vector Machine (SVM) [31,36,37], was considered because it achieved good results in related works.

Therefore, this work covered the analysis of a set of six different algorithms to identify which is the best for each user, based on the F1 Score (F1), accuracy (AC), and the complexity of the algorithm: (a) RF; (b) SVM; (c) XGB; (d) GB; and (e) NBB; and (f) NBG, for continuous authentication, both static and dynamic. As an additional checkpoint, the SVM One-Class algorithm was included for the location pattern. The tools used for building the authentication models were the scikit-learn library and the XGBoost python.

To adjust the hyperparameters of the algorithms based on ensemble methods and SVM, the Grid Search technique was used, using a predefined parameter list for the RF, SVM, GB, and XGB algorithms, as shown in Table 3.

### 3.5. Evaluation Metrics

In this version of the framework, the main evaluation metrics are F1 Score (F1) and FAR. F1 [38,39,40] is the harmonic mean between Recall and Precision [38], and its definition and meaning can be found in [40,41]. This metric was chosen instead of precision, as this new version of the framework has multiclass scopes, so if precision is used, the true negatives have a high weight, which could generate good accuracy but does not necessarily indicate good model performance.

Regarding FAR [42,43,44,45], it is used to measure the performance of the model regarding impostors that are accepted as legitimate users.

### 3.6. Fusion of Scores

For the proposed model, the fusion rule for the classifier results is the score average of accuracy for the location pattern, and F1 for Moments 1 and 2, as shown in Figure 5. Opposite to [11], which proposed to deduct the standard deviation from the mean value, this requirement has been removed in this new version of the framework because the new model also performs a test to distinguish an impostor from all other users of the system. Therefore, the penalty generated in the final score by deducting the standard deviation is unnecessary since all score values are only accepted if they have a value of more than 90% for both the location and the Moments 1 and 2. Moments 1 and 2 use the F1 of the best classifier, which can be individualized for the user or shared with others, but always consider F1 for the whole class.

## 4. Results and Discussion

This section presents the experimental scenarios developed for the validation of the new authentication framework and discusses the results of the validation process.

### 4.1. Experimental Scenarios

Three scenarios (S1, S2, and S3) were defined to specify the minimum number of interactions with the application required to create a supervised machine learning model that can obtain a 90% F1 Score (F1) and a maximum FAR of 10%. Each user model is confronted with the templates from other users, as shown in Table 4, based on the number of templates that the user generated during the experimental period. A ratio of one login template to three interactions with the application is used in the test scenarios since, to complete a flow, the user must interact with three screens after login: (1) Services Menu screen, (2) Transaction Menu screen, and (3) Transaction screen.

As detailed in Table 4, for the user to participate in SV within scenario 1 (S1), they would need at least 10 login interactions with the application. Out of these generated templates, at least 5 would be used for the training phase, and all other user-generated templates would be used for testing. In scenario 2 (S2), the user should have at least 15 templates, 10 of which would be used for training while all others would be used for testing. To participate in scenario 3 (S3), there would have to be at least 20 interactions with the login screen, 15 of which would be used for training and all others for testing. For the user to participate for DV in scenario 1 (S1), the user would need at least 10 post login interactions with the application generating 30 templates, out of which at least 15 would be used for the training phase and all the other user-generated templates would be used for testing. In scenario 2 (S2), the user should have used at least 45 templates, where 30 would be used for training and all others for testing. To participate in scenario 3 (S3), there would have to be at least 60 templates, 45 of which of these would be used for training and all others for testing.

### 4.2. Users Templates

For the detailed scenarios of the 51 users who installed the application, only 25 participated in the experiment as legitimate users, since only these have generated the minimum number of templates for S1: 10 interactions with the login and at least 10 complete interactions in the transaction flow, representing 30 post login templates. The data of the other 26 users who did not provide the minimum number of templates needed for scenario 1 were used to generate imposter data for training and tests and were also used to calculate the imposters’ FAR. The details of the 25 legitimate users are presented in Table 5, as are the number of templates that each user generated during the experiment.

Based on the maximum number of templates generated by the 25 legitimate users, the number of users who would participate in each scenario and each moment were defined according to Table 6.

### 4.3. Model Implementation

To build the authentication model according to the framework rules, the following steps were carried out:All six supervised machine learning algorithms are trained and tested with balanced data. The same number of vectors, lines contained in the templates, is used for legitimate and illegitimate users based on the number of vectors contained in the legitimate user templates;The algorithms that obtained F1 from 90% are identified;Those models that obtain 100% accuracy are then discarded because this behavior may indicate overfitting or that the data is not yet sufficient to define the user pattern;If among the models with F1 at 90% there is NBB or NBG, they are preferred as they are simpler and faster algorithms for prediction. Otherwise, the model with the highest F1 value is selected;The best authentication model selected in the previous step is then confronted with data from at least 50 other users. The model is only considered good if it obtains a maximum I_FAR of 10%, which in this case may represent that, among the 50 other imposter users, 5 have a behavior pattern that is identified as similar to that of the evaluated user, based on features used in the experiment.

### 4.4. FeaturesRanking

To understand the representativeness of each feature for the creation of the models and how they could influence the creation of the scopes, a ranking of the feature was created using the RF algorithm with multiple classes, which is detailed in Table 7 (top 10 features for SV) and in Table 8 (top 10 features for DV).

From these rankings, it can be observed that the features with the greatest importance for both SV and VD are finger size and average finger size. These two features have a weight close to or greater than 40% for the definition of the models when using the multiclass scope. This behavior served as a basis for the creation of experiment scopes, seeking to understand how the models behave without these features. The different test and training scopes were created based on the ranking of the features, so the scopes with the best performance could be evaluated for the proposed framework by capturing features that are relevant to less training and testing time.

### 4.5. Scopes

For each scenario, S1 to S3, the framework was trained and tested in six different scopes in order to find the scopes with the best performance, respecting the number of templates for training and testing of each scenario, as detailed in Section 4.1. Each scope was created based on features, and if the model is multiclass or binary, the scopes are characterized as follows:aScope A (SA): using all features captured in Moments 1 and 2 and generating one model per user;bScope B (SB): excluding features related to sensor data and generating one model per user;cScope C (SC): excluding the finger size and average finger size features because these features are identified with a high weight in features ranking and generating one model per user;dScope D (SD): using all features captured in Moments 1 and 2 and generating only one model for all users;eScope E (SE): excluding features related to the sensor data and generating only one model for all users;fScope F (SF): excluding the finger size and average finger size features and generating only one model for all users.

SA is the same scope used in [11] while the others were created for this new framework. The idea was to find the best performance of the authentication framework for the collected data and the creation of the models in the scenarios of the experiment.

Given the results captured in the static and dynamic verification scopes from SA to SF, as can be seen in the Section 4.6.1 and Section 4.6.2 for SV and in Section 4.7.1 and Section 4.7.2 for DV, it was defined that the framework will incorporate scopes SD, SA, and SB, in this order.

SD takes precedence over other scopes because, in SD, the model has already been tested with data from all of the other users that generated the model in addition to this approach providing a reduction in the number of possible overfittings that may happen in individual models that were only trained with two classes, the legitimate users (1) and the imposters (0), since in a shared model the internal classes can have an accuracy of up to 100% without necessarily overfitting the model. Additionally, it was noticed that SA could be complementary to SD.

Scopes SA and SB are only used if it is not possible to find an F1 of at least 90% for the user in SD. If an F1 of at least 90% is not found for the class in SD, then training in SA is carried out. If it is still not possible to find the defined F1, training in SB is carried out. The ineffectiveness in finding the expected F1 in any of the scenarios may be an indication that the user usage pattern cannot be captured with the proposed system and scenarios. The description of the algorithm is described in Algorithm 1.

### 4.6. Experimental Results for SV

In this subsection, the results are analyzed for SV between the proposed scopes and scenarios. In the following tables, the ALG field indicates the best algorithm, the ALG(S) field indicates the algorithm and scope together, the QTD field indicates the total number of templates for the user and, the I_FAR field indicates the FAR of the model about the authentication of the templates of all the other 50 users imposters. The light-gray lines indicate that the model also met the requirement of Step 5, to have a FAR less than 10%, that is, it was possible to find a model that met all of the requirements of the framework in one or more of the scenarios.

**Algorithm 1:** The flow of the framework algorithm with scopes.

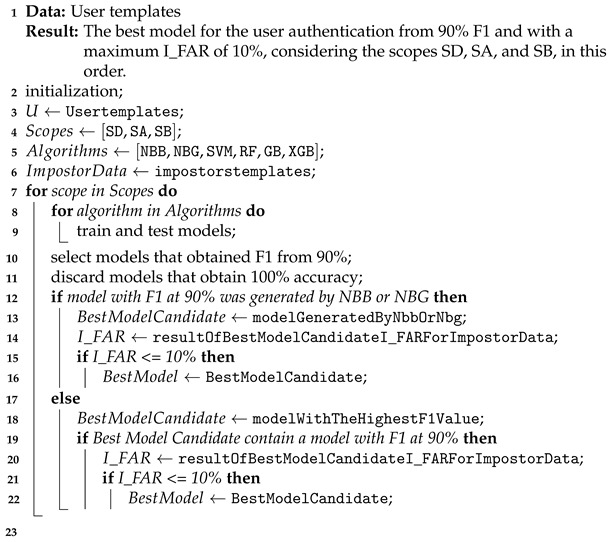



#### 4.6.1. Results for SV between Scenarios

With the proposed approach, for SV, uniting the SA, SB, and SD scopes in the framework, it was possible to find an algorithm with the F1 intended for 92% of users in one or more scenarios, which was not found by only two users. Scope D was responsible for solving the search for 52% of users in S1, 66% in S2, and 62.23% in S3; Scope A was responsible for 32% of users in S1, 16.66% in S2, and 15.38% in S3; and the Scope B was responsible for 4% of users in S1, 0% in S2, and 0% in S3, thus indicating that Scope D is comprehensive in finding the best algorithm for the data collected and analyzed in this experiment for static verification.

The results for Step 5 of the proposed framework, the impostor FAR test, are detailed in Table 9. This is the last screening that the model must pass to be considered suitable for user authentication.

Based on the steps defined for the framework, it was possible to find a model with an F1 of at least 90% and with an I_FAR of up to 10% for 80% of users in SV, of which five out of five users offered enough samples to participate in S1, indicating that the use of the proposed SV framework, if used in conjunction with conventional methods such as passwords, can offer an additional line of security with good performance. In the case of this experiment, in a test environment with different devices and different scopes, it provided a quality model for 20 of the 25 users, which is the majority of the users who participated in the experiment.

#### 4.6.2. Average Results for SV

In this subsection, the results of the proposed framework are analyzed in terms of the expected results concerning the results observed in the work listed in the literature review. The average results presented in Table 10, Table 11 and Table 12 were calculated using the simple average between the values found for each user. All comparisons are made so that the proposed framework can have a minimum benchmark for performance comparison, even though the features and metrics used in each literature-reviewed work are not the same as those used in the proposed framework. The general results of the framework and the scopes that compose it, SA, SB, and SD, are marked in bold.

Regarding the average results for SV, according to Table 10, it was possible to obtain a result of up to 98.25% accuracy with the model proposed in the framework in scenario 3, even if this is not the metric considered in the proposed model. For comparison purposes, a greater accuracy than the one described in the literature review was obtained, where the highest reported accuracy values were 96% for static verification in a mobile banking application [18] and 93.04% for static verification in [7]. Based on Table 11, which shows the average EER, the values varied between 4.57 and 1.88%, and F1, which is detailed in Table 12 was resulted between 95.32 and 97.05%, with the average results above the threshold defined in 90%, indicating that the proposed model managed to obtain a good performance for most users who participated in the SV experiment.

### 4.7. Experimental Results for DV

In this subsection, the results are analyzed for DV between the proposed scopes and scenarios. In the following tables, the ALG field indicates the best algorithm, the ALG(S) field indicates the algorithm and scope together, the QTD field indicates the total number of templates for the user, and the I_FAR field indicates the FAR of the model for authentication of the templates of the other 50 users or imposters. The light-gray lines indicate that the model also met the requirement of step 5: to obtain a FAR less than 10%. It was possible to find a model that met all of the requirements of the framework in one or more of the scenarios.

#### 4.7.1. Results for DV

The proposed approach with DV as well as the SA, SB, and SD scopes for the framework made it possible to find, an algorithm with F1 intended for 86.95% users in one or more scenarios, leaving only four users, of which three did not offer enough samples to participate in scenario 3 without a model with an F1 intended for dynamic verification. Scope D was responsible for solving for 17% of users in S1, 27% in S2, and 81.81% in S3; Scope A was responsible for solving for 13.04% of users in S1, 38.88% in S2, and 9.09% in S3; and Scope B was responsible for solving for 30.43% of users in S1, 16.66% in S2, and 0% in S3. Unlike static verification, where the resolution of the intended F1 search was concentrated in SD, in the case of dynamic verification, there was a greater distribution among the scopes, indicating that the task of finding a model with the intended F1 for DV is more complex, which involves a set of screens, some that have only one touch on the screen for interaction, and not just one screen, as in SV.

The result for Step 5 of the proposed framework, the imposter FAR test, is detailed in Table 13. This is the last screening that the model must pass to be considered suitable for user authentication.

Based on the steps defined in the framework, it was possible to find a model with an F1 of at least 90% and with an I_FAR of up to 10% for 69.56% of users in DV (16 users); for 7 of them, it was not possible to obtain a model that met the requirements: users must provide enough samples to participate in up to S2. Of the users who provided enough samples to participate in S3, only one did not have a good model created, indicating that, for DV, more training data are needed to build a model with good performance. Therefore, it was suggested that the use of post-login interaction data for user authentication is promising.

#### 4.7.2. Average Results for DV

In this subsection, the results of the proposed framework are analyzed in terms of the expected results concerning the results observed in the work listed in the literature review. The average results presented in Table 14, Table 15 and Table 16 were calculated using the simple average between the values found for each user. All comparisons were made so that the proposed framework can have a minimum benchmark for performance comparison, even though the features and metrics used in each literature-reviewed work are not the same as those used in the proposed framework. The general results of the framework and the scopes that compose it, SA, SB, and SD, are marked in bold.

As detailed in Table 14, for DV, the accuracy varied between 90.1 and 98%. In Table 15, it was possible to observe an EER of up to 3.07% in scenario 3, lower than that reported in the literature for dynamic authentication in [8], with 4% between weeks, and as shown in Table 16, the F1 score varied between 90.68 and 95.72, an average value higher than the threshold defined at 90%, indicating that the proposed model performed well for most users.

### 4.8. Algorithm Frequency

During the experiments and as explained before, six different algorithms were used to create the models between the scenarios that would meet all of the requirements defined for the framework. The algorithms that met all of the proposed framework’s requirements varied, as shown in Table 17. In the case of SV, the best algorithms were RF, with higher frequencies, followed by NBG and NBB. For DV, the algorithm with the highest frequency was NBG, followed by RF.

The results in Table 17 show that the best algorithms varied based on ensemble and on Naive Bayes, with the best results obtained for RF in SV and for NBG in DV. It was also possible to verify that the SVM algorithm, despite being referenced in the literature as having good results in [7], did not obtain good results in our experiments or in the creation of models that met all of the requirements of the framework with GB and XGB. It is possible to note that the issue addressed in the experiment, with the features used, can be solved using simpler algorithms, such as NBB, NBG, and RF.

### 4.9. Outlier Detection

A model to be used in behavioral biometrics has to be good at classifying a legitimate user, maintaining a balanced FAR and False Rejection Rate (FRR) at low rates. This indicates that the model is effective in both identifying legitimate users and imposters.

As detailed in Section 4.7.2, the EER varied between 1.88 and 4.57% for SV and between 3.07 and 9.85% for DV, keeping it balanced and with low values, especially when considering only S3, with 1.88% for SV and 3.07% for DV, between the two-week duration of the test.

Besides this balance and the good values found, the model tries to make it more resilient to imposters. In this regard, only the model that obtained an F1 larger than 90% and a FAR lower than 10% were considered and used. The obtained values were then confronted with the templates of all of the other 50 users who participated in the experiment.

Models that passed all of the steps defined by the framework and had I_FARs greater than 1% did not identify an impostor as a 100% legitimate user in any of the cases evaluated. An I_FAR value greater than 1% was the sum of authentications, within the threshold of acceptance, of some data vectors for different imposters authenticated as legitimate users.

The proposed framework can be highly efficient in identifying impostors (at least 90% F1) based on the proposed features and scopes when using a mobile banking application.

Even with all of the precautions defined in the model creation requirements for the framework, an imposter can still have access to the system without being detected, as there are margins of error, and as shown in [46], the data from continuous authentication based on behavioral biometrics can suffer imitation attacks.

In [46], data from the various sensors available on smartphones were not taken into account, unlike the experiment proposed in this work. For attacks against touch behavioral biometrics, considering sensor data, the authors in [47] suggested that the consideration of sensor data can be a strong biometric authentication mechanism against recently proposed practical attacks.

### 4.10. Fused Results

The last step in generating the final score in the proposed framework is to combine the values generated for accuracy of the location with the F1 Score values generated for Moments 1 and 2 by calculating the arithmetic mean of these values. To represent the result of the fusion of scores according to the proposed model, the first result was selected between the scenarios for SV and DV. It considered the users who had a model created for Moments 1 and 2 according to the requirements of the framework. The value of the location accuracy taken into account is always the scenario with the highest number according to the SV and DV results.

This is just an example, as the results of SV and DV were obtained with the authentication of several templates from the same user. However, in a real scenario, the fusion of scores comes from the results related to only one template at the run time in SV, DV, and location. The results for the fusion of scores is shown in Table 18.

According to the results demonstrated in this subsection, the proposed framework was able to find satisfactory models for SV and DV for 14 of the 25 users who participated in the experiment. For SV, the majority were satisfactory in scenario 1; for DV, there was a greater distribution among the scenarios, indicating that, in general, more data are needed in DV than in SV to create a quality model.

### 4.11. Comparison with Previous Work

In this subsection, we compare the results observed in the literature review with our continuous authentication framework in Table 19.

Based on the observations made in the literature review, which are summarized in Table 19, the continuous authentication framework proposed in this work, developed for continuous authentication based on touch dynamics biometrics, and focused on mobile banking applications presents better results than that in [19], which proposed models that also use continuous authentication based on both static and dynamic verification for authentication of the user, during the entire interaction with an application.

Figure 6 illustrates the performance results achieved by the continuous authentication framework proposed in comparison with the best results from the reviewed literature, as detailed in Table 19, for accuracy and F1.

As detailed in the Figure 6, our accuracy results were better than those reported in [7,8,18]. The proposed framework took into account the F1 score to avoid the strong influence that accuracy can have on true negatives. On the other hand, it also considered the accuracy metric as well, which was also better than the other related works.

Figure 7 illustrates the performance results achieved by the continuous authentication framework proposed in comparison with the best results from the reviewed literature, as detailed in Table 19, for EER.

If compared with [19], our model achieved an EER of 1.88%, which is much better than the 11.5% found in [19]. Our proposed approach showed superior results concerning the work presented in the literature review focusing on mobile banking applications and addresses both static and dynamic verification.

Therefore, in addition to the capture layers when typing a password and interactions with the application, the proposed model also captures of information from different sensors that are present in mobile devices, such as the rotation and acceleration sensors, and from the user location pattern, such as that used in [16,20], respectively.

Besides the search for an algorithm that has an F1 of at least 90%, the framework proposes an additional step so that the model is accepted within the defined requirements: it must be confronted using data generated by all other users who participated in the experiment and cannot have an I_FAR greater than 10%. This step was incorporated with the objective of finding balanced models regarding FAR and FRR when using balanced training and test data and when tested with imposter data to make the models more resilient and robust.

Regarding the identification of impostors, the approach of using this extra verification step makes it more difficult for a fraudster to be able to impersonate a legitimate user when using the application since, in addition to obtaining their password, the impostor must have a pattern of interaction with the application quite similar to that of the legitimate user in the face of a model that has already been tested against the data of at least 50 imposters.

## 5. Conclusions and Future Work

The experimental results obtained during this research reinforce our thesis on the complexity of finding a machine learning algorithm that is suitable for several different users since the pattern of interaction with an application is unique to each individual. The use of six different algorithms by the proposed framework appears to be a promising approach to overcoming this difficulty.

To validate the framework as an additional method used against identity-related fraud, three different scenarios were created. The scenarios were based on the number of templates generated for testing and training in order to understand how the models would perform in each case evaluated. Six different scopes were also created based on removing some features to understand how they influence the performance of the model and with different types of model (multiclass or binary).

For construction of the models according to the experiments carried out with the six different machine learning algorithms, the F1 score was used. Due to the scopes being multiclass and binary, there was a need for a general metric in which the true negative was not weighted as high.

A relevant characteristic involved in the steps of creating the model was verification of the FAR for imposters, making the models good and more resilient since they were tested once more with the data of all other users who participated in the experiment.

Regarding the scopes studied in this work, the ones that presented the best results were SA, SB, and SD. These scopes were incorporated into the new framework, leading to good results. In SV, it was possible to find a satisfactory model for 20 of the 25 users who participated in the experiment. In DV, a suitable model for 16 users out of 23 was found. It was also observed that SA could be complementary to SD.

For the investigation into the creation of more robust models with the inclusion of FAR verification of imposters, it was possible to observe that this is an important step as the models need to be balanced between the identification of imposters and legitimate users, with low rates of FRR and FAR.

To be considered good, all models, when trained with balanced data, had to pass the test with only up to 10% FAR for imposters when confronted with data from all other users of the experiment. Therefore, the inclusion of the I_FAR step enriched the framework and made the models more robust.

The algorithms that showed the best performance had results that varied between the scopes. Regarding SV, a better result was observed for RF, an algorithm based on the ensemble method, followed by the algorithms based on Naive Bayes, NBG, and NBB. In the case of DV, NBG is the best performing algorithm, with a higher frequency, followed by NBG.

For the average F1 and EER, the results varied between 90.68% and 97.05% and between 9.85% and 1.88%, respectively, among the scenarios from the proposed scopes. Therefore, it validates the promising perspective of the use of touch dynamics biometrics as a new layer of security if used in conjunction with traditional methods such as passwords. Such layers can evolve in combination as security layers to mitigate authentication fraud in mobile banking applications.

### Future Work

Future work related to this research includes field studies to capture data in a real online banking application with a larger number of users and a longer duration to validate the proposed framework in a setup closer to the final application.

Additionally, an important question to consider is the relationship between cell phone models and the quality of the data coming from the capture sensors, yielding an interesting analysis on how device quality reflects the quality of the model of the device user generated.

Regarding the performance of the proposed framework, promising perspectives in this investigation include considering methods based on adaptive selection by weighting features of interest and using advanced feature engineering techniques. Furthermore, the application of filters on the data captured from the sensors possibly improves the performance of the models created.

Since continuous authentication is an evolving field, further studies on appropriate and better-performing machine learning algorithms, and better understanding of the impacts of their parameters and the related approaches for data collection and preservation, model training, deployment, and maintenance are also needed.

A related study will consider analyzing and comparing the performance and cost of continuous authentication methods based on physiological and behavioral biometrics.

Finally, as the proposed framework needs to show resilience against attacks, future studies must conduct further exploration of adversarial models to implement the necessary countermeasures.

## Figures and Tables

**Figure 1 sensors-21-04212-f001:**
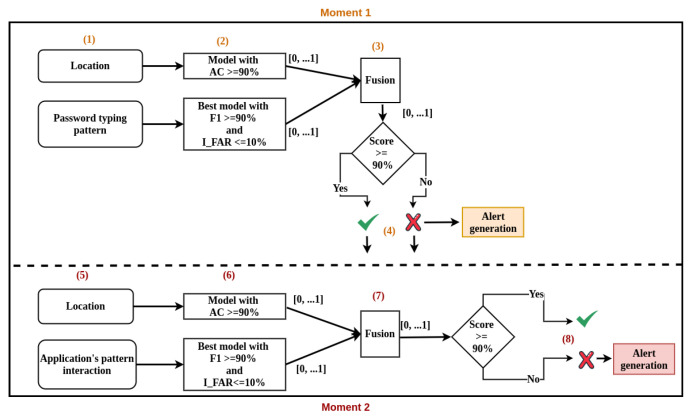
Proposed framework.

**Figure 2 sensors-21-04212-f002:**
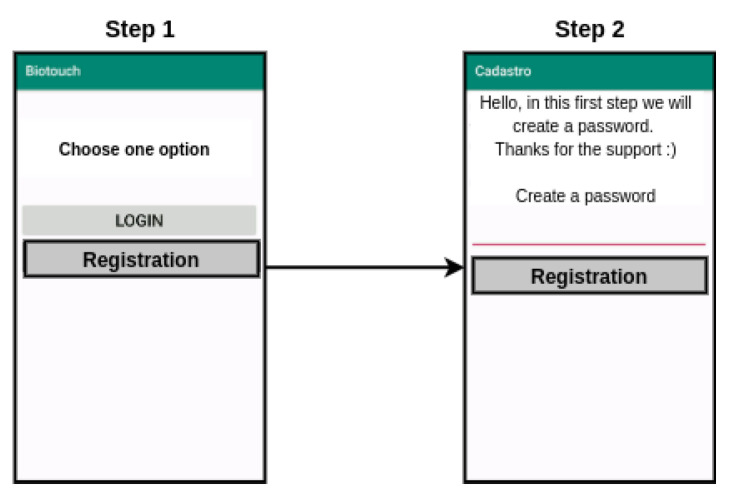
Biotouch app screens for the registration flow.

**Figure 3 sensors-21-04212-f003:**
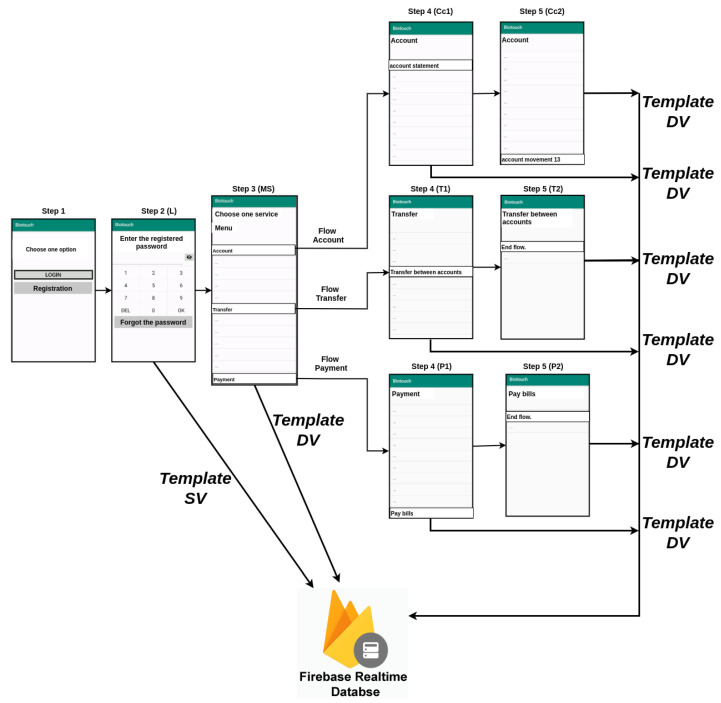
Biotouch app screens for the application interactions.

**Figure 4 sensors-21-04212-f004:**
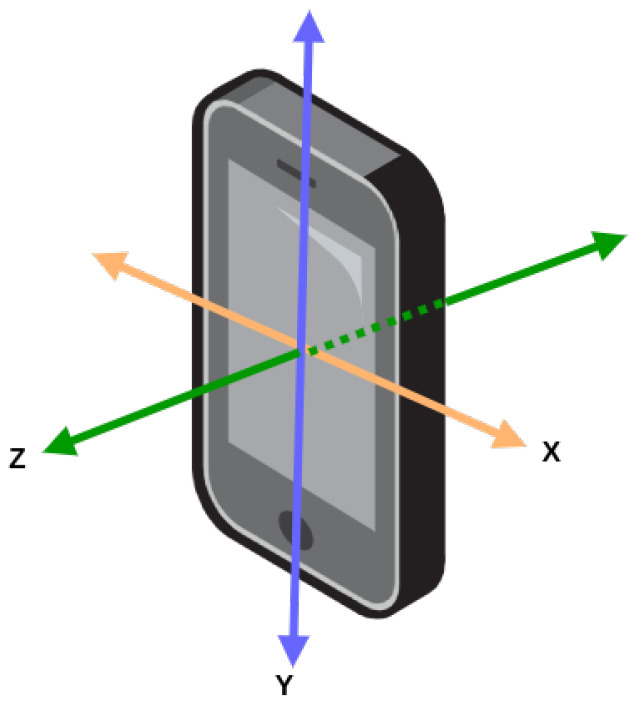
Representation of a smartphone axes.

**Figure 5 sensors-21-04212-f005:**
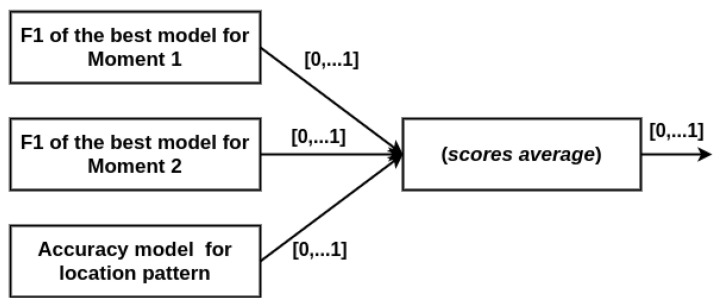
Diagram for the fusion of results.

**Figure 6 sensors-21-04212-f006:**
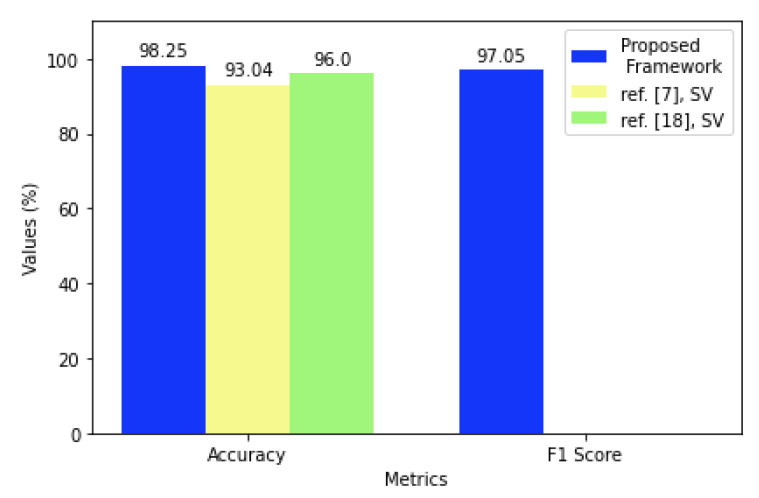
Performance comparison of the proposed framework with the best results from the reviewed literature for AC and F1.

**Figure 7 sensors-21-04212-f007:**
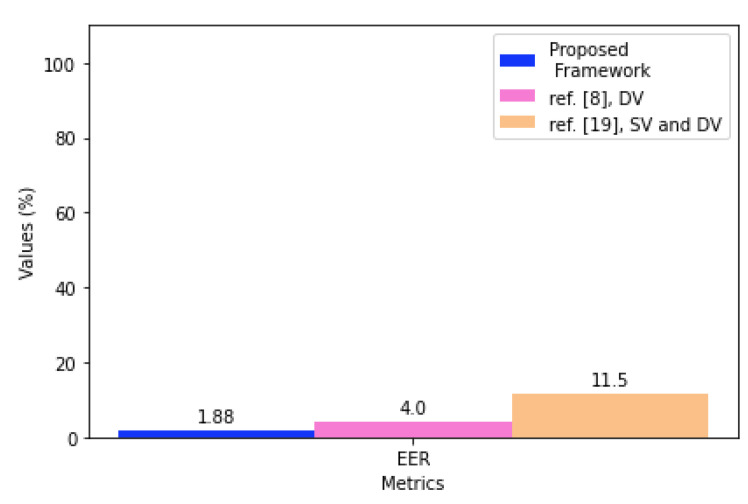
Performance comparison of the proposed framework with the best results from the reviewed literature for EER.

**Table 1 sensors-21-04212-t001:** Features collected for Moments 1 and 2.

Feature	Sensor
Down Down Time	Touchscreen
Down Up Time	Touchscreen
Up Down Time	Touchscreen
Up Up Time	Touchscreen
Average Down Up Time	Touchscreen
Pressure	Touchscreen
Average Pressure	Touchscreen
Figer Size	Touchscreen
Average Finger Size	Touchscreen
Acceleration force along the X axis (including gravity) [25]	Accelerometer
Acceleration force along the Y axis (including gravity) [25].	Accelerometer
Acceleration force along the Z axis (including gravity) [25].	Accelerometer
Rate of rotation around the X axis [25].	Gyroscope
Rate of rotation around the Y axis [25].	Gyroscope
Rate of rotation around the Z axis [25].	Gyroscope
Geomagnetic field of the environment for the physical X axis in μT [26].	Magnetometer
Geomagnetic field of the environment for the physical Y axis in μT [26]	Magnetometer
Geomagnetic field of the environment for the physical Z axis in μT [26]	Magnetometer
Rotation vector component along the X axis (X * sin(θ/2)) [25]. (software or hardware)	Rotation Sensors
Rotation vector component along the Y axis (Y * sin(θ/2)) [25]. (software or hardware)	Rotation Sensor
Rotation vector component along the Z axis (Z * sin(θ/2)) [25]. (software or hardware)	Rotation Sensor
Scalar component of the rotation vector ((cos(θ/2)) [25]. (software or hardware)	Rotation Sensor
Estimated heading Accuracy [27]. (software or hardware)	Rotation Sensor
Acceleration force along the X axis (excluding gravity) [25]. (software or hardware)	Acceleration Sensors
Acceleration force along the Y axis (excluding gravity) [25]. (software or hardware)	Acceleration Sensors
Acceleration force along the Z axis (excluding gravity) [25]. (software or hardware)	Acceleration Sensors
Force of gravity along the X axis [25]. (software or hardware)	Gravity Sensors
Force of gravity along the Y axis [25]. (software or hardware)	Gravity Sensors
Force of gravity along the Z axis [25]. (software or hardware)	Gravity Sensors

**Table 2 sensors-21-04212-t002:** Hyperparameters of the RF algorithm for definition of the feature ranking.

Name	Value
n_estimators	40
n_jobs	2
random_state	0
bootstrap	False
criterion	entropy
max_depth	5
max_features	9
min_samples_leaf	3

**Table 3 sensors-21-04212-t003:** Hiperparameters used for algorithms based on ensembles and SVM.

Algorithm	Name	Value
**SVM**	kernel	rbf, linear, poly
	gamma	scale, auto, $1\times 10^{-2}$, $1\times 10^{-3}$, $1\times 10^{-4}$
	C	0.0000001, 0.000001, 0.00001, 0.0001, 0.001, 0.1
	class_weight	{0:1,1:2}, balanced
	degree	3, 5
**RF**	random_state	0
	n_jobs	2
	n_estimators	20, 25, 30
	max_depth	3, 5, None
	max_features	1, 3, 5, auto
	min_samples_leaf	0.3, 0.4, 0.5
	min_samples_split	0.3, 0.4, 0.5, 6
	bootstrap	True, False
	criterion	gini, entropy
	class_weight	{0:1,1:2}, balanced
**GB**	n_estimators	10, 20, 30, 75, 100
	learning_rate	0.001, 0.01, 0.1
	max_depth	5, 6, 7
	min_samples_split	0.3, 0.4, 0.45, 0.5
	min_samples_leaf	0.20, 0.25, 0.3, 0.4
	max_features	3, 7, 10, 20, None, balanced
**XGB**	n_estimators	20, 30, 40, 100
	colsample_bytree	0.6, 0.7, 0.8
	max_depth	15, 20, 25
	reg_alpha	1.1, 1.2, 1.3
	reg_lambda	1.1, 1.2, 1.3
	subsample	0.7, 0.8, 0.9

**Table 4 sensors-21-04212-t004:** Details of the experimental scenarios.

Type	Number of Templates	Number of Templates for Train	Number of Templates for Test	Scenario
**SV**	From 10	5	From 5	1
	From 15	10	From 5	2
	From 20	15	From 5	3
**Type**	**Number of Templates**	**Number of Templates for Train**	**Number of Templates for Test**	**Scenario**
**SV**	From 10	5	From 5	1
	From 15	10	From 5	2
	From 20	15	From 5	3
**DV**	From 30	15	From 15	1
	From 45	30	From 15	2
	From 60	45	From 15	3

**Table 5 sensors-21-04212-t005:** Number of templates of the 25 legitimate users participating in the experiment.

User	Number of Templates per Screen								
**Identification**	**L**	**MS**	**Cc1**	**Cc2**	**T1**	**T2**	**P1**	**P2**	**Total**
drds94uTXlk	12	8	6	6	1	1	1	1	36
cwdNwCqtFAs	11	10	3	3	4	4	3	3	41
eHC7qNMAdCI	10	11	3	3	5	5	2	2	41
dUbEbDq40fM	12	12	5	5	3	3	4	4	48
fUB30EtiU0Q	12	12	4	4	5	5	3	3	48
cWN_XjnNRDw	14	14	5	5	3	3	6	6	56
ffdzWINClJ4	14	16	6	4	6	6	4	4	60
dFOtPe4f8Xc	18	22	8	7	7	7	7	7	65
cBc3b9Cv4X0	17	17	5	5	5	5	7	7	68
ev9fChXnR3I	17	18	11	11	2	2	5	5	69
fRA_pBm0ks4	18	18	6	6	6	6	6	6	72
dyRTk2BUAeo	21	19	6	6	6	6	5	5	74
cFxPtyX-07w	18	20	8	7	7	7	4	4	75
dVGmimRO7bE	20	19	5	5	9	9	5	5	77
fGTU-LDm8uM	20	20	7	7	6	6	7	7	80
fmVXDwdw20Q	25	21	7	7	7	7	7	7	88
eqmPzjjzrHk	23	25	5	4	11	10	9	9	96
enJGMKaiFiI	27	26	7	7	13	13	6	6	107
fIewI06H8u8	31	30	10	10	10	10	10	10	121
eerZKZFi2kY	40	40	12	12	15	15	13	13	160
dlj7Igoq3HQ	53	54	18	17	17	17	19	19	214
fDAVsmA3HUY	69	70	22	22	24	24	24	24	279
f0ttzpoZyeA	75	75	13	13	52	52	10	10	300
eoWIhgawcZ0	84	85	35	35	26	26	23	23	337
fhw9jzhvkGs	90	90	38	38	49	49	3	3	360

**L:** login; **MS:** application menu service screen; **Cc:** application account screen; **Cc1:** application account menu screen; **Cc2:** application account transaction screen; **P:** application payment screen; **P1:** application payment menu screen; **P2:** application payment transaction screen; **T:** application transfer screen; **T1:** application transfer menu screen; **T2:** application transfer transaction screen.

**Table 6 sensors-21-04212-t006:** Number of participants per scenario SV and DV.

Participants SV e DV			
	S1	S2	S3
**SV**	25 of 25	18 of 25	13 of 25
**DV**	23 of 23	18 of 23	11 of 23

**Table 7 sensors-21-04212-t007:** Top 10 ranking features for Moment 1.

Feature	Importance Value Moment 1
**Característica**	**Importance Value Moment 1**
Figer Size	0.240485
Average Finger Size	0.236985
Pressure	0.065277
Average Pressure	0.060130
Geomagnetic field on Y	0.058717
Geomagnetic field on X	0.053064
Geomagnetic field on Z	0.044218
Average Down Up Time	0.042234
Scalar component of the rotation vector	0.036942
Acceleration force along the Y axis (including gravity)	0.031548

**Table 8 sensors-21-04212-t008:** Top 10 ranking features for Moment 2.

Feature	Importance Value Moment 2
Average Finger Size	0.268114
Finger Size	0.138374
Scalar component of the rotation vector	0.090839
Pressure	0.090122
Average Pressure	0.087559
Geomagnetic field on Y	0.056308
Geomagnetic field on X	0.051553
Rotation vector component along of X	0.034605
Acceleration force along the Y axis (including gravity)	0.030806
Geomagnetic field on Z	0.025307

**Table 9 sensors-21-04212-t009:** Final results for SV framework scenarios.

User	SV Framework Final Result								
**Identification**	**ALG (S) S1**	**F1 S1**	**I_FAR S1**	**ALG (S) S2**	**F1 S2**	**I_FAR S2**	**ALG (S) S3**	**F1 S3**	**I_FAR S3**
drds94uTXlk	RF (SD)	94.23	0	—	—	—	—	—	—
cwdNwCqtFAs	NBG (SA)	99.32	0	—	—	—	—	—	—
eHC7qNMAdCI	RF (SD)	100	0.6	—	—	—	—	—	—
dUbEbDq40fM	NBB (SA)	94.33	6.08	—	—	—	—	—	—
fUB30EtiU0Q	GB (SA)	98.51	44.41	—	—	—	—	—	—
cWN_XjnNRDw	RF (SD)	100	13.93	—	—	—	—	—	—
ffdzWINClJ4	RF (SD)	95.84	0	—	—	—	—	—	—
dFOtPe4f8Xc	NBG (SB)	93.24	11.15	—	—	—	—	—	—
cBc3b9Cv4X0	NBG (SA)	98.75	4.1	NBG (SA)	97.54	0	—	—	—
ev9fChXnR3I	—	—	—	RF (SD)	95	1.08	—	—	—
fRA_pBm0ks4	RF (SD)	100	1.51	RF (SD)	100	3.72	—	—	—
dyRTk2BUAeo	RF (SD)	98.84	0.14	RF (SD)	94.88	0.37	RF (SD)	100	0.03
cFxPtyX-07w	RF (SD)	99.27	0.09	RF (SD)	100	0.4	—	—	—
dVGmimRO7bE	RF GB XGB (SA)	90.64	77.75	RF (SD)	99.78	0.06	RF (SD)	100	0.03
fGTU-LDm8uM	RF (SD)	91.34	0.09	RF (SD)	93.89	0.05	RF (SD)	99.71	0.03
fmVXDwdw20Q	NBB (SA)	95.47	6.66	RF (SD)	100	0.01	RF (SD)	99.08	0.02
eqmPzjjzrHk	RF (SD)	100	1.62	RF (SD)	100	1.69	RF (SD)	100	1.42
enJGMKaiFiI	RF (SD)	97.85	0.2	RF (SD)	95.29	0.29	RF (SD)	99.77	0.64
fIewI06H8u8	RF (SD)	99.11	2.35	RF (SD)	99.49	0.29	RF (SD)	100	1.32
eerZKZFi2kY	—	—	—	—	—	—	—	—	—
dlj7Igoq3HQ	@c@NBB (SA)	99.91	9.06	NBB (SA)	99.54	10.35	RF (SD)	95.72	0.58
fDAVsmA3HUY	@c@NBG (SA)	96.87	0	NBG (SA)	96.64	8.69	NBG (SA)	96.26	8.85
f0ttzpoZyeA	RF (SD)	97.03	0.78	RF (SD)	98.2	1.9	RF (SD)	97.92	2.05
eoWIhgawcZ0	RF (SD)	99.16	2.29	RF (SD)	100	0.62	RF (SD)	100	2.14
fhw9jzhvkGs	—	—	—	—	—	—	—	—	—

**ALG(S) S1:** Algorithm (Scope) Scenario 1, **F1 S1:** F1 Score Scenario 1, **I_FAR S1:** Impostors FAR Scenario 1, **ALG(S) S2:** Algorithm (Scope) Scenario 2, **F1 S2:** F1 Score Scenario 2, **I_FAR S2:** Impostors FAR Scenario 2, **ALG(S) S3:** Algorithm(Scope) Scenario 3, **F1 S3:** F1 Score Scenario 3, **I_FAR S3:** Impostors FAR Scenario 3. **Note:** The light-gray lines indicate that the model also met the requirement of Step 5, to have a FAR less than 10%, that is, it was possible to find a model that met all of the requirements of the framework in one or more of the scenarios.

**Table 10 sensors-21-04212-t010:** Average accuracy per scenario SV.

Average Accuracy Per Scenario SV							
**Scenario**	**SA**	**SB**	**SC**	**SD**	**SE**	**SF**	**Framework**
**Cenário**	**SA**	**SB**	**SC**	**SD**	**SE**	**SF**	**Framework**
**S1**	**89.44**	**87.12**	86.64	**98.85**	96.82	97.61	**95.81**
**S2**	**91.02**	**88.35**	84.7	**98.13**	95.87	97.05	**97.38**
**S3**	**90.77**	**87.29**	87.55	**98.33**	96.06	97.96	**98.25**

**Table 11 sensors-21-04212-t011:** Average EER per scenario SV.

Average EER per scenario SV							
**Scenario**	**SA**	**SB**	**SC**	**SD**	**SE**	**SF**	**Framework**
**Average EER per scenario SV**							
**Cenário**	**EA**	**EB**	**EC**	**ED**	**EE**	**EF**	**Framework**
**S1**	**10.14**	**11.45**	13.51	**7.3**	19.18	11.95	**4.57**
**S2**	**8.97**	**11.62**	15.23	**5.87**	18.47	11.47	**2.87**
**S3**	**9.21**	**12.63**	12.49	**4.59**	13.23	6.46	**1.88**

**Table 12 sensors-21-04212-t012:** Average F1 per scenario SV.

Average F1 per scenario SV							
**Scenario**	**SA**	**SB**	**SC**	**SD**	**SE**	**SF**	**Framework**
**S1**	**90.4**	**88.24**	87.91	**83.79**	59.59	70.58	**95.32**
**S2**	**91.68**	**87.32**	86.9	**85.34**	60.53	73.91	**96.34**
**S3**	**91.44**	**83.7**	87.55	**87.9**	71.62	84.58	**97.05**

**Table 13 sensors-21-04212-t013:** Final results for the DV framework scenario.

User	DV Framework Final Result								
**Identification**	**ALG (S) S1**	**F1 S1**	**I_FAR S1**	**ALG (S) S2**	**F1 S2**	**I_FAR S2**	**ALG (S) S3**	**F1 S3**	**I_FAR S3**
cwdNwCqtFAs	NBG (SB)	96.6	0	—	—	—	—	—	—
dUbEbDq40fM	RF (SD)	94.25	0.01	—	—	—	—	—	—
fUB30EtiU0Q	XGB (SA)	99.5	53.64	—	—	—	—	—	—
cWN_XjnNRDw	—	—	—	—	—	—	—	—	—
ffdzWINClJ4	—	—	—	—	—	—	—	—	—
dFOtPe4f8Xc	NBG (SB)	99.1	0	RF (SB)	98.23	11.01	—	—	—
cBc3b9Cv4X0	XGB (SB)	99.92	11.16	NBG (SB)	99.84	15.16	—	—	—
ev9fChXnR3I	—	—	—	—	—	—	—	—	—
fRA_pBm0ks4	RF (SD)	98.77	10.13	RF (SD)	100	5.11	—	—	—
dyRTk2BUAeo	—	—	—	SVM (SA)	99.85	14.87	—	—	—
cFxPtyX-07w	RF (SD)	96.54	0.2	RF (SD)	100	0.14	—	—	—
dVGmimRO7bE	NBG (SB)	91.87	0	RF (SD)	100	0.5	—	—	—
fGTU-LDm8uM	—	—	—	—	—	—	NBG (SD)	99.37	0
fmVXDwdw20Q	NBG (SB)	95.37	8.71	SVM (SA)	98.97	25.38	SVM (SA)	99.48	23.41
eqmPzjjzrHk	NBB (SA)	99.92	32.13	RF (SD)	100	0.75	NBG (SD)	99.46	100
enJGMKaiFiI	—	—	—	XGB (SA)	90.4	52.6	NBG (SD)	90.2	0
fIewI06H8u8	—	—	—	—	—	—	NBG (SD)	92.07	0
eerZKZFi2kY	—	—	—	—	—	—	—	—	—
dlj7Igoq3HQ	NBG (SB)	98.29	8.27	GB (SA)	92.28	65.9	NBG (SD)	96.72	0
fDAVsmA3HUY	NBG (SB)	99.94	0	NBG (SA)	97.19	0	NBG (SD)	91.68	0
f0ttzpoZyeA	NBG (SA)	91.51	59.03	NBG (SA)	90.23	0	NBG (SD)	100	0
eoWIhgawcZ0	RF (SD)	98.72	0.17	RF (SD)	98.42	1.22	NBG (SD)	99.97	0
fhw9jzhvkGs	—	—	—	RF (SA)	90.79	72.04	NBG (SD)	98.36	0

**ALG(S) S1:** Algorithm(Scope) Scenario 1, **F1 S1:** F1 Score Scenario 1, **I_FAR S1:** Impostors FAR Scenario 1, **ALG(S) S2:** Algorithm(Scope) Scenario 2, **F1 S2:** F1 Score Scenario 2, **I_FAR S2:** Impostors FAR Scenario 2, **ALG(S) S3:** Algorithm(Scope) Scenario 3, **F1 S3:** F1 Score Scenario 3, **I_FAR S3:** Impostors FAR Scenario 3. **Note:** The light-gray lines indicate that the model also met the requirement of Step 5, to have a FAR less than 10%, that is, it was possible to find a model that met all of the requirements of the framework in one or more of the scenarios.

**Table 14 sensors-21-04212-t014:** Average accuracy per scenario DV.

Average Accuracy Per Scenario DV							
**Scenario**	**SA**	**SB**	**SC**	**SD**	**SE**	**SF**	**Framework**
**S1**	**80.29**	**85.72**	80.27	**96.89**	96.5	96.32	**90.1**
**S2**	**84.38**	**85.22**	81.84	**95.49**	95.22	97.01	**93.34**
**S3**	**84.86**	**90.88**	80.76	**99.21**	95.77	97.22	**98**

**Table 15 sensors-21-04212-t015:** Average EER per scenario DV.

Average EER Per Scenario DV							
**Scenario**	**SA**	**SB**	**SC**	**SD**	**SE**	**SF**	**Framework**
**S1**	**21.15**	**14.31**	19.71	**21.63**	25.9	26.55	**9.85**
**S2**	**15.82**	**14.73**	18.12	**11.51**	25.24	19.14	**6.61**
**S3**	**15.15**	**9.09**	19.24	**2.69**	21.1	11.84	**3.07**

**Table 16 sensors-21-04212-t016:** Average F1 per scenario DV.

Average F1 Per Scenario DV							
**Scenario**	**SA**	**SB**	**SC**	**SD**	**SE**	**SF**	**Framework**
**S1**	**81.71**	**87.09**	81.95	**55.18**	47.92	50.17	**90.68**
**S2**	**87.32**	**88.43**	85.43	**75.78**	48.21	60.14	**93.73**
**S3**	**88.07**	**90.35**	85.13	**93.61**	60.19	73.89	**95.72**

**Table 17 sensors-21-04212-t017:** Framework algorithm frequencies.

	SV			DV		
	**S1**	**S2**	**S3**	**S1**	**S2**	**S3**
**NBB**	3	1	—	—	—	—
**NBG**	3	2	1	6	2	8
**SVM**	—	—	—	—	—	—
**RF**	12	12	10	3	5	—
**GB**	—	—	—	—	—	—
**XGB**	—	—	—	—	—	—

**Table 18 sensors-21-04212-t018:** Fusion of the SV, DV, and location results.

User	Fusion of *Scores* Per User					
**Identification**	**SV Scenario**	**SV F1**	**DV Scenario**	**DV F1**	**Location AC**	**Fusion Result**
cwdNwCqtFAs	1	99.32	1	96.6	90.62	95.51
dUbEbDq40fM	1	94.33	1	94.25	92.3	93.62
fRA_pBm0ks4r	1	100	2	100	92.82	97.6
cFxPtyX-07w	1	100	2	100	92.82	97.6
dVGmimRO7bE	2	99.78	1	96.54	95.83	97.21
fGTU-LDm8uM	1	91.34	3	99.37	92.76	94.49
fmVXDwdw20Q	1	95.47	1	95.37	92.61	94.48
eqmPzjjzrHk	1	100	2	100	93.19	97.73
enJGMKaiFiI	1	97.85	3	90.2	89.47	92.5
fIewI06H8u8	1	99.11	3	92.07	92.5	94.56
dlj7Igoq3HQ	1	99.91	1	98.29	92.18	96.79
fDAVsmA3HUY	1	96.87	1	99.94	92.62	96.47
f0ttzpoZyeA	1	97.03	2	90.23	92.63	93.29
eoWIhgawcZ0	1	99.16	1	98.72	92.95	96.94

**Table 19 sensors-21-04212-t019:** Comparative summary of the characteristics of the proposed framework and related work.

Work	[7]	[8]	[18]	[19]	This Proposed Framework
**Statical Verification**	x	—	x	x	x
**Dynamic Verification**	—	x	—	x	x
**Sensors**	Touchscreen	Touchscreen	Touchscreen, Accelerometer, Orientation, Gravity, Magnetometer, Gyroscope	Touchscreen, Accelerometer	Touchscreen, Accelerometer, Orientation, Gravity, Magnetometer, Gyroscope, Rotation, Acceleration
**Undetermined Devices**	—	—	x	—	x
**Number of Users**	42	41	95	22	25
**Number of Algorithms**	7	2	3	1	7
**Best Result**	93.04% accuracy	0% to 4% EER inter-week	96% accuracy	11.5% EER	97.05% F1 *Score*, 1.88% EER, 98.25% accuracy
**Algorithm with best Result**	RF	SVM	RF	*Fuzzy*	RF, NBG

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
