# Peer review of "A Framework for Continuous Authentication Based on Touch Dynamics Biometrics for Mobile Banking Applications"

_sensors, 2021, doi:10.3390/s21124212_

Round 1
Reviewer 1 Report
The group of authors proposed previously a supervised Machine Learning based framework for continuous authentication using multiple scopes called Biotouch. In the reviewed paper the authors report the performance improvement of the Biotouch framework and the creation of more resilient models for it, by using multiple scopes and testing the models for the False Acceptance Error (FAR) of the imposter. Indeed, they consider:
- A multiple scope approach, so that the authentication models are validated for different feature sets, with the best performing scopes being added to the framework;
- Six different scopes were developed to improve the performance;
- With the selected multiple scopes, the efficiency metric of the models present a minimum F1 score of 90%.
The improvements of the continuous authentication framework are introduced in full detail in a clear and readable way, and the experimental performance is seemingly fair.
Finding a machine learning algorithm that is suitable to be considered by several different users is a hard work since the pattern of interaction with an application is unique to each individual. The proposed use of six different algorithms appeared to be a promising and realistic approach to overcome this difficulty.
Author Response
We would like to thank the reviewer for the evaluation of our work.
Reviewer 2 Report
This paper presents a framework for user continuous authentication for mobile banking applications using touch dynamic biometrics. The manuscript is well-structured and tackles an interesting problem with several research merits. However, a few comments should be taken into consideration to improve the quality and the information presentation within the manuscript.
- Since the study is about continuous authentication using machine learning, I am not sure about the validity of using the term framework in the title and within the text. A framework can be a type of guideline that other users/researchers can use to build something, which is not the goal of this paper. Additionally, Biotouch is a mobile App and not a framework. Furthermore, do mobile banking applications need continuous authentication? I strongly believe, and as a mobile banking user, that mobile banking applications require one-shot (or physiological biometrics) authentication as the user is not continuously open and interacts with the mobile banking Apps.
- In the abstract, the authors claim to propose an improved Biotouch framework is one of the contributions. However, such improvements and the drawbacks of the previous Biotouch version are unclear. Furthermore, the abstract neglects the value of the physiological biometric authentication methods, which I think the abstract should balance between the two types of authentication approaches. Finally, the achieved performance should be briefly compared to the performance of other methods in the state-of-the-art.
- The data collection section mentions 29 features were collected via a mobile application. However, the section does not say any information about the number of persons who have used this application. It is very important to have enough information about the collected dataset to be able to judge the performance of the proposed model. Actually, not having sufficient instances in the dataset may make the results unconvincing.
- Table 2 is not readable, so please consider reformatting this table.
- I would like to see more information about the design/creation of the ML model(s). I strongly recommend that the manuscript includes data in an algorithmic format, so other researchers can follow up and the model reproducibility will increase. The current information about the model creation is succinct.
- Section 4.2 states that the collected data includes 51 users in total. It means that the dataset includes only 51 instances and for some scenarios, it includes 25 users/instances. Regardless of that this information should be explained in the data collection section, the size of the dataset is very small, and hence, the performance of the proposed model is unreliable. It is not acceptable to train and evaluate a proposed model using a dataset of only 25 or even 51 users.
- The authors should calculate, and the manuscript should present a picture of the correlation between the ranked features to make sure there is no overfitting in the proposed model. Furthermore, the purpose of the feature ranking process is unclear to me. Was the goal to reduce the number of features? Was to goal to reduce the training/testing time? Furthermore, what is the goal of the scoping process?
- The paper should present a comprehensive comparative study with other methods in the literature. According to Figure 6, I cannot see a significant improvement in the identification accuracy compared to Ref. 7 and Ref. 17. The authors should consider that Ref. 17 has almost double users (97) compared to the proposed method, which makes the results in Ref. 17 more reliable. The authors may consider other parameters like the number of features and the processing time as part of the comparison.
- After reading the whole manuscript, I could not identify a linkage between the proposed method and the mobile banking applications. I do believe that the scope and the title of the manuscript should be revised to reflect the actual applications of the proposed method.
- The manuscript should be checked by an English native speaker to catch and correct any typo or grammatical error like the one in Table 1 caption.
Author Response
We would like to thank the reviewer for the careful review of the previous work.
We are uploading (a) our point-by-point response to the comments (below) (response to reviewers)

Reviewer 3 Report
(1)The abstract is too long. Some introduction on the background can be removed or shortened.
(2)Fusion and feature selection are employed in this paper. The review of the related works and comparison experiments can be more sufficient. Please carefully read, cite and compare (if applicable) the following papers that are closely related to fusion and feature selection.
Adaptive selection/weighting of features is typically used for dimensionality reduction and performance improvement. The features with high discrimination [-] and low correlation [--] should be selected for fusion and provided with high weights.
-Dynamic weighted discrimination power analysis: a novel approach for face and palmprint recognition in DCT domain. International Journal of the Physical Sciences
--PalmHash Code vs. PalmPhasor Code
Fusion has several advantages, which can be performed at image level, feature level, score level, decision level. Image-level fusion requires image registration that is not always available. Feature-level fusion can reduce the information leakage (i.e. strong security) [-], but suffers from the incompatibilities of the dimensionality and type between the source features. Score-level fusion and decision-level fusion avoid the incompatibilities [--], but they suffer from the information leakage problem (i.e. weak security), high computation complexity in matching stage and high storage cost for several templates.
-Dual-source discrimination power analysis for multi-instance contactless palmprint recognition
--Dual-key-binding cancelable palmprint cryptosystem for palmprint protection and information security
If the authors cannot employ these methods or compare their method with these methods, at least they could introduce/mention these novel technologies in Sections 1 and 2 to improve the quality of the survey, or explain them as the possible future works.
Author Response
We also would like to thank the reviewers for the careful review of the previous work.
We are uploading our point-by-point response to the comments (below) (response to reviewers).
Best regards,
The authors.

Reviewer 4 Report
The authors introduce a framework for continuous person authentication based on touch dynamics. The concept and results presented in the paper have immediate practical applications and can solve real-world problems especially for mobile banking applications.
The literature survey is comprehensive and the provided references are recent and relevant.
The proposed framework is clearly presented. I consider the experimental results to be correct and relevant. The comparison with other algorithms/methods is well presented.
My concern is related to the security of the stored templates. Is there any method to increase their security? Is it possible to temper the template stored in the device memory or transmitted to Firebase database? I recommend the authors to discuss the security of the solution in a paragraph in the paper.
Author Response
We also would like to thank the reviewer for the careful review of the previous work.
We are uploading our point-by-point response to the comments (below) (response to reviewers).
Best regards,
The authors.

Round 2
Reviewer 2 Report
Thank you for fulfilling my comments. I can still see that the manuscript needs to highlight the relevance of the study to the banking systems, or otherwise, the authors should make/claim that the proposed approach is generic and can be used with other application domains. I see that the comparison to the state-of-the-art is still weak. I am not sure the reason that the author has modified Table 6 and created Table 7 ( as mentioned in the response letter) to address my comment about building a comparative analysis. I am not convinced about the number of features used in this study.
Author Response
Dear Reviewer,
We also would like to thank the reviewer for the careful review of the previous work. We believe this new version have greatly improved the quality of the manuscript based on the reviewer’s comments.
We are uploading our point-by-point response to the comments (below)
Best regards,
The authors.

Reviewer 3 Report
The current version is better, but there are still some typos.
E.g.
-The order of the references is confusing. [11] follows [9]. ... Please reorder the references according to their citation sequence in the text.
- fusion of features, scoring level fusion, decision level fusion [13] and image-level [14]. (Line 161)
-> image-level fusion, scoring-level fusion [*], decision-level fusion [13] and feature-level fusion [14].
*PalmHash Code vs. PalmPhasor Code
-The surname of the 1st author of [14] is wrong. “T” should be deleted.
...
The authors should carefully check the article.
Author Response
Dear Reviewer 3,
We would like to thank the reviewer for the careful review of the previous work. We believe this new version have greatly improved the quality of the manuscript based on the reviewer’s comments.
We are uploading our point-by-point response to the comments (below).
Best regards,
The authors.
